# OpenReview forum: "End-to-end Invariance Learning with Relational Inductive Biases in Multi-Object Robotic Manipulation"
_ICLR.cc/2023/Conference — Submitted to ICLR 2023_

### Official Review · Reviewer_E3aJ · 2022-10-20

**Confidence:** 4
**Correctness:** 4
**Technical Novelty And Significance:** 3
**Empirical Novelty And Significance:** 2
**Recommendation:** 5

**Clarity, Quality, Novelty And Reproducibility:**

The paper is well-written, and the proposed method is novel to the best of my knowledge.
The authors do not release code, but they do provide many details in the main text and appendix of the paper.

**Strength And Weaknesses:**

Strengths:
1. The proposed method is clearly motivated. The authors seek to encode strong inductive biases to enable out-of-distribution generalization.
2. The experiments conducted are thorough. The authors compare their method against a few baselines on two manipulation tasks, and show that the trained policies extrapolate to situations beyond what they are trained on.

Weaknesses:
1. I can't help but feel that the empirical results are not very impressive. In both manipulation tasks, the relational reasoning boils down to "ignore the distractor objects", and the generalization amounts to increasing the number of distractor objects beyond what the policy was trained it. Are there other, more diverse tasks that better showcase the compositional generalization that this approach should enable?

**Summary Of The Paper:**

The authors propose a variant of relational networks to reason about interactions between objects and between objects and an agent in a manipulation task. They train policies with this architecture on a few simulated manipulation tasks, and demonstrate out-of-distribution generalization to variants of the task not seen during training.

**Summary Of The Review:**

The approach seems sound, but the experiments feel a bit lackluster. I would not be upset if this paper were accepted, but I also don't feel particularly excited about it.

---

> ### Author Response · Authors · 2022-11-18
> **Response to Reviewer E3aJ**
>
> We thank the reviewer for their comments and would like to address some of the concerns in the following:
>
> It is correct that the final policy in our experiments boils down to correctly learning to “ignore the distractor objects”. As we show in the paper, the only baseline currently found in the literature does not manage to learn such an invariant policy, and therefore it cannot generalize to a different number of objects than what was observed at training time. In our opinion, this is raising a new concern for the community. In particular one of our main contributions is to highlight that attention-based approaches are showing limitations in large-scale robotic multi-object manipulation applications where it is not feasible to observe that many objects at training time.

---

### Official Review · Reviewer_vRaR · 2022-10-22

**Confidence:** 2
**Clarity, Quality, Novelty And Reproducibility:** The LRN appraoch seems novel
**Correctness:** 2
**Technical Novelty And Significance:** 2
**Empirical Novelty And Significance:** 2
**Recommendation:** 6

**Strength And Weaknesses:**

Strengths: The paper is clearly written and the results show a clear improvement over the compared methods.

Weaknesses: The RN framewor, which is used as a basis here, could have been explained in more detail. It is hard to reproduce the entire algorithm only based on equation (1).

**Summary Of The Paper:**

The paper presents an RL-approach for multi-object manipulation. The main idea is to extrapolate a learned skill to manipulate one object in a zero-shot fashion onto a number of additional objects. To do this, the authors propose an exension of the recent Relation Network RN by Santtoro et al. In simulated experiments, the authors show that their approach improves over RN and an attention-based module.

**Summary Of The Review:**

Interesting approach, but some more background is needed for the non-expert reader to assess the novelty.

---

> ### Author Response · Authors · 2022-11-18
> **Response to Reviewer vRaR**
>
> We thank the reviewer for their comments and feedback that allowed us to improve our work further. We added a table (Table 2) in our appendix describing in detail our implementation of the Linear Relation Network to complement the information provided in equation (1).

---

### Official Review · Reviewer_T4pA · 2022-10-24

**Confidence:** 4
**Correctness:** 3
**Technical Novelty And Significance:** 2
**Empirical Novelty And Significance:** 3
**Recommendation:** 3

**Clarity, Quality, Novelty And Reproducibility:**

* The paper applies relational networks borrowed from the question answering line of literature, and claims that they are not studied in the context of RL. Hence the study, to me, is more empirically than technically novel.

* Some of the presentation aspects of the work can be improved (*W2-5*)

**Strength And Weaknesses:**

**Strengths**

*S1.* The method is simple to implement and performant in the two tasks considered.

*S2.* Experiments generally back the main claims of the paper related to generalization to new numbers of distractor cubes.

**Weaknesses**

*W1.* I am not convinced of the fidelity of the simulation environment. In Fig. 1 it appears that the red and yellow blocks are overlapping, which is not physically realistic. This calls into question the results. It seems possible that some agents are able to exploit such artifacts in the simulation environment. I suggest doing a more thorough investigation of the simulation environment and increasing the collision checking in pybullet to prevent such cases.

*W2.* My understanding is that the trends in Fig. 2 (left) are extrapolated for the blue and green lines after x=1 and x=2 respectively. This could be misleading to a reader who looks at the figure at a glance. It is unclear if there is enough data to extrapolate these trends. I recommend running these experiments rather than extrapolating the trends.

*W3.* Clarity: some of the task/experimental details are not provided. For example, how are the states of each object $s_i$ represented? As 2D coordinates, 3D coordinates, poses? What are the inputs and outputs to the networks?

*W4.* What is the intuition on why LRN is able to generalize to different numbers of objects but the baselines cannot? Making this more clear in the manuscript could help guide the readers towards the key takeaways.

*W5.* What are the small object specific MLPs? Is there one for each cube? If so how do you decide which MLP to use for each cube at test time when there are more cubes? Is it based on color? Please consider including more details in this section.

*W6.* The method uses GT state information.

*W7.* The method, while effective for the downstream tasks, is potentially lacking in generality. How would the method perform if shapes that are not cubes are included, such as YCB objects? What about if state was inferred? Are there other tasks that this method could be applied to?

*W8.* The paper argues that the network could be learning higher level interactions in deeper layers of the network. This suggests another baselines: only tokens are fed into $g(.)$ without any pairing. This may further elucidate the effects of the inductive bias considered in the paper.

**Minor**

*M1.* In the abstract, I recommend rewording the phrase: "allows agents to extrapolate and generalize zero-shot to any new object number." This reads as an unsupported claim as it is not empirically verifiable and there is no proof included about why this is guaranteed.

*M2.* I recommend removing phrases like "horrendous amounts of data," which feel a bit editorial and imprecise.

*M3.* I suggest removing the phrase "we suspect that this generalization capability could possibly even hold beyond the 9-distractor environments." Alternatively, I suggest pushing the method with even more objects to see at what point it can no longer handle more distractor objects.

**Summary Of The Paper:**

The paper considers two block manipulation tasks, where the number of train and test block are different. They (1) demonstrates the computation and performance degradation of graph attention and relational networks as more blocks are introduced at test time and (2) introduce a linear relation network module that empirically mitigates the problems mentioned in the aforementioned baselines.

**Summary Of The Review:**

The main claims of the paper surrounding are validated (i.e., the proposed method can generalize to completing target manipulation tasks with more distractor objects than seen during training). Additionally the presentation in Fig. 4 and 5, which some some of the main empirical results, are well executed.

However, I currently recommend rejection of the manuscript for the following reasons: key details necessary to understand the method are missing (*W3-5*). The experiments currently seem limited to simple settings, further experimentation seems necessary to more rigorously verify the method and determine its impact (*W6-8*). Additionally, Fig. 2 is misleading as it contains extrapolation, which may not be obvious to a reader who is glancing at the paper (*W2*).

POST REBUTTAL:
changing score from 5 to 3. See "Post rebuttal" for more details.

---

> ### Author Response · Authors · 2022-11-18
> **Response to Reviewer T4pA**
>
> We thank the reviewer for their comments and would like to address some of the concerns in the following:
>
> W1. The red block in figure 1 is just a visualization of the goal position where we would like to push the cube of the same color and therefore it is not a physical object in pybullet. There is therefore no technical issue with the simulation environment using pybullet, the red block is only shown for visualization purposes in the paper.
>
> W2. Those curves represent the zero-shot capabilities of our modules (RN and LRN). This means that we are not extrapolating them but rather showing that there is no need to train longer than what’s necessary for x=1 and x=2 as we can just use those models without any further training also with many distractors
>
> W3. Table 1 in the appendix describes in detail what is included as the state for each type of object.
>
> W5. In our setup, there are 3 different types of objects (goal, cubes, and robotic fingers) which are computed from states of different sizes as provided by the CausalWorld environment. For this reason, we have 3 object-specific MLPs which map these different types of objects to an embedding of the same size
>
> W6. Solving tasks on this particular robotic platform is already challenging from GT state information. Given that even with GT state information on the hardware we used, training a single seed for 1 billion timesteps took 2 weeks, training tasks from pixels would be unfeasible and outside of the scope of the project. In addition, we are not aware of prior works that can learn manipulation tasks on this platform end-to-end from pixels.
>
> W7.  In this work, we are interested in generalizing to the number of objects we did not experiment with different shapes as this is an unrelated form of generalization regarding the objective we are interested in solving. The lack of generalization of currently used architectures renders training unfeasible for a larger number of objects, as we show in figure 2.
>
> W8. We would like to kindly point out that this suggested method is actually already covered and presented in fig. 6 (appendix) as “LRN 2D No Rel”. (we considered it an ablation in the context of our work as we could not find this method being presented in earlier work)

---

> > ### Comment · Reviewer_T4pA · 2022-11-20
> > **Post rebuttal**
> >
> > Thank you to the authors for their response. However, after reviewing the responses and the comments of other reviewers, I have elected to decrease my score from a 5 to a 3. My main concerns and justification for the rating are as follows:
> >
> > * use of GT state information
> > * limited task setting making it unclear how useful the proposed architecture is more broadly
> > * lack of clarity in the manuscript
> >
> > I believe that the work would greatly benefit from addressing the above three points.

---

### Official Review · Reviewer_Pmgk · 2022-10-24

**Confidence:** 4
**Correctness:** 3
**Technical Novelty And Significance:** 2
**Empirical Novelty And Significance:** 2
**Recommendation:** 3

**Clarity, Quality, Novelty And Reproducibility:**

+ The paper is clearly written.
- The contribution, which is essentially a simplifying assumption in the representation of the state, is not novel.

**Strength And Weaknesses:**

+ Good justification for the problem of generalizing to a variable number of distractors in robotics manipulation.
- The proposed setting is extremely simple: manipulating individual objects, in simulation, under the assumption of perfect knowledge. The authors assume that the resulting policy will be so simple that it can be represented by a "small MLP" that doesn't even need specifying.
- The contribution is essentially limited to formula (1) for the representation. This formula essentially amounts to ignoring all the relations except the binary relations ones between the goal cube and the individual distracting cubes. While this is indeed O(n), it is not a novel contribution, only a simplifying assumption.

**Summary Of The Paper:**

The paper is considering the problem of learning robotics manipulation in a multi-object setting. The specific problem considered is the successive repositioning of two cubes by a manipulator in the presence of a variable number of distractor cubes.

The authors argue that previous approaches, based on GNNs do not learn policies that generalize to a variable number of distractors. They proposed a representation that is based on a relation network, and show that this approach can extrapolate to a variable number of distractors. The authors also propose a "linearized relation network" module which works in a linear rather than a quadratic computational complexity.

**Summary Of The Review:**

The paper considers the case of a robotics manipulation of a cube in the presence of distractor cubes in a simulation setting. The contribution of the paper is a simplified relation network based representation where only the relations of the goal cube is considered. The paper shows that with this representation the learned policy generalizes better to variable number of distractors.

---

I read the authors answer to my and the other reviewers comments. The answers do not change my rating of the paper.

---

> ### Author Response · Authors · 2022-11-18
> **Response to Reviewer Pmgk**
>
> We thank the reviewer for their comments and would like to address some of the concerns in the following:
>
> First, we would like to point out that our main contributions are not only restricted to the LRN module. In this work, we are studying a so-far underexplored problem setup where the number of cubes used at train time is fixed and generalization must be induced at an architectural level to avoid the increased complexity arising when directly training on a large number of objects.
>
> Furthermore, describing our change to the RN as an assumption is not correct as we do not decrease its ability to reason about any pairwise relation and therefore our change is not limiting RN’s generality (we discuss in the appendix how reducing the number of relations computed does not actually affect the ability to reason about higher-level relations not computed explicitly by the LRN module).

---

### Official Review · Reviewer_4jkK · 2022-10-25

**Confidence:** 4
**Correctness:** 2
**Technical Novelty And Significance:** 2
**Empirical Novelty And Significance:** 2
**Recommendation:** 3

**Clarity, Quality, Novelty And Reproducibility:**

* Clarity: high.

The paper is clearly-written.

* Quality: medium.

The experiment can be performed in a more systematic manner.

* Novelty: mid-low.

Arguably the proposed architecture has a mediocre technical novelty.

* Reproducibility: high.

The proposed idea can be implemented straightforwardly and the appendix delivers details.

**Strength And Weaknesses:**

**Strength**
* The paper is easy to read and well structured
* The scope of problem is clearly defined
* The proposed idea is simple and intuitive

**Weaknesses**
* Scalability: the proposed method seems limited to the proposed task
* The baselines are limited
* Task setting (or scope of the problem) is limited


**Summary Of The Paper:**

The paper presents a simple relational architecture that can learn and generalize on multi-object manipulation tasks. The main benefit of the proposed method is the linear-scale complexity in terms of the number of objects (i.e., distractors) in the single-goal tasks. The proposed method is tested on robotic manipulation tasks and compared against a pairwise relational model and an attention-based model. The empirical results demonstrate that the proposed linearized relation network (LRN) module improves the computational complexity and extrapolates to an unseen number of objects.

**Summary Of The Review:**

Overall, I have three main concerns on the significance of this work.

*1. Single-object goal task setting is limited*

This work assumes the task always involves a single object. In section 2, authors claim that the proposed task formulation can be seen as a generic subtask, so that composing such subtasks can complete a bigger task involving multiple objects. However, each subtask may still involve more than one object. Imagine a subtask of navigating to a certain location. While navigating,  the robot should not break any objects that may be useful for later subtasks or harm any person. Also if the robot is carrying a glass of water, it should try not to spill the water while navigating. The robot may need to choose a safer path depending on the state of the “glass of water” object (which is not the goal object). Given that the proposed task setting is quite limited, the strict dependence of the proposed method on the task setting further limits the significance of this work (continued in the next section).

*2. The proposed method is limited to the proposed task setting*

The main contribution of this work comes from the linearized relational module. And such linearization would not harm the performance only on a single-goal task, where relations between other objects are not or less important. As discussed above, relation between other objects may be as important as goal objects. In section 5.5, authors discuss that the higher-order relations can be captured by the MLP policy head. However, this means the proposed LRN cannot handle the higher-order relations alone, which makes the main claim less convincing.

*3. Baselines are limited*

As discussed in the previous point, the proposed method exploits the inductive bias (i.e., goal-conditioned task where goal is based on single object) in the task design. However, it seems the baselines do not benefit from such specific task design. The proposed method is compared with two baselines: attention-based model and pairwise relational model. However, both models do not have any inductive bias regarding the goal-based task, hence it is not a fair comparison. A more fair baseline can be appending the goal token to every paired input tokens in the pairwise relational model (i.e., goal + obj1 + obj2 instead of obj1 + obj2). Another important baseline to compare with is the set representation (e.g., [1]) with goal awareness (e.g., appending a goal token before/after the aggregation). On the other hand, there are many works on linearizing the quadratic architectural complexity such as LinFormer [2]. Comparing with the goal-aware version of linearized attention-based or relational models can strongly support the main claim.

[1] Zaheer et al., "Deep Sets", 2017

[2] Wang et al., "Linformer: Self-attention with linear complexity", 2020

---

> ### Author Response · Authors · 2022-11-18
> **Response to Reviewer 4jkK**
>
> We thank the reviewer for their comments and would like to address some of the concerns in the following:
>
> 1. "Single-object goal task setting is limited":
>
> Multi-object manipulation tasks we are interested in, given the current state of the robotic manipulation literature, do not involve complex objects such as a “glass of water” but rather where the complexity comes from a large number of objects. A common example is stacking cubes, for which our single-goal task can be seen naturally as a subtask.
>
> 2. "The proposed method is limited to the proposed task setting":
>
> First, we would like to point out that our main contributions are not restricted to the LRN module. In this work, we are studying a so-far underexplored problem setup where the number of cubes used at train time is fixed and generalization must be induced at an architectural level to avoid the increased complexity arising when directly training on a large number of objects.
>
> The LRN is used to move from a variable-size vector to a fixed-sized one, its role is to “compress” the state into a representation that is still expressive enough to allow a policy to pick the right action. Therefore, its goal is to represent the multi-object information such that a simple MLP policy can interpret it properly, which means encoding relations depending on the information they carry.
>
> 3. "Baselines are limited":
>
> Each of our baselines includes the goal token as one of the objects, if this was not the case the baselines would not be able to solve even the training task as the goal position is randomly sampled. The attention network is the reimplementation of what is widely used in the literature [1,2,3,4,5], whereas when adapting the RN we made novel design choices. Examples are: the introduction of the Normalization layer to handle the change in the number of objects at test time, and the choice of not conditioning on the goal (similar to what is done in [6] when conditioning on the question) since in early experiments using goal+obj1+obj2 showed inferior performances with respect to our current version and we stopped experimenting with that design.
>
> The Deep Set approach is a general approach that, in the invariant regime of Theorem 2, proposes to use sum as an aggregation function over some representation $\phi(x)$ of each element x of the set X. In figure 6 (appendix), we consider as an ablation the setup where no pairwise relations are computed, and therefore closely reproducing what is proposed in the Deep Sets paper.
>
> The LinFormer architecture, and the related line of research of bringing transformer architectures to linear complexity, were not taken into account, as these approaches are mostly studied and developed to reduce inference and training time but not for achieving better performances either at train or test time.
>
> [1]: Vinicius Zambaldi, David Raposo, Adam Santoro, Victor Bapst, Yujia Li, Igor Babuschkin, Karl Tuyls, David Reichert, Timothy Lillicrap, Edward Lockhart, Murray Shanahan, Victoria Langston, Razvan Pascanu, Matthew Botvinick, Oriol Vinyals, and Peter Battaglia. Relational deep reinforcement learning, 2018.
>
> [2]: Richard Li, Allan Jabri, Trevor Darrell, and Pulkit Agrawal. Towards practical multi-object manipulation using relational reinforcement learning, 2019.
>
> [3]: Andrii Zadaianchuk, Maximilian Seitzer, and Georg Martius. Self-supervised visual reinforcement learning with object-centric representations, 2020.
>
> [4]: Matthew Wilson and Tucker Hermans. Learning to manipulate object collections using grounded state representations, 2020.
>
> [5]: Allan Zhou, Vikash Kumar, Chelsea Finn, and Aravind Rajeswaran. Policy architectures for compositional generalization in control. 2022.
>
> [6]: Adam Santoro, David Raposo, David G. T. Barrett, Mateusz Malinowski, Razvan Pascanu, Peter Battaglia, and Timothy Lillicrap. A simple neural network module for relational reasoning, 2017.

---

> > ### Comment · Reviewer_4jkK · 2022-11-22
> > **Re: Response to Reviewer 4jkK**
> >
> > Thank you for the detailed responses to my comments.
> >
> > 1. "Single-object goal task setting is limited"
> > I agree that the proposed single-object goal task setting is not too limited for conventional robotic object manipulation tasks scope. However, current direction is making more domain-specific restrictions for performance gain, where the 'domain' may not be too broad enough. So, I agree with the authors response, but also still have some concerns regarding significance of this work.
> >
> > 2. "The proposed method is limited to the proposed task setting"
> > I also acknowledge the contribution in promoting the overlooked challenge in object manipulation. My main concern was in methodological contribution. Connected to 1, I believe the proposed inductive bias is "you can pay less attention to the relation between non-goal objects", and I believe this is too restrictive, which makes the proposed method hard to scale beyond "conventional" object manipulation tasks.
> >
> > 3. "baseline is limited"
> > First, I was not assuming that the baseline does not take the goal token or not (although this can be made more clear in the paper).
> > And it is great that authors have already compared with goal+obj1+obj2 setting. It would be helpful to show the comparison in the paper or in appendix. If the computation cost is the problem, I believe only showing the early learning trend should suffice assuming a significant performance difference. Regarding Deepset, it's helpful to see a similar baseline in Appendix and authors might want to consider presenting the Figure 6 result in main text. Also, I still think it would be helpful to try the exact version of Deepset, because I believe the specific architecture proposed in Deepset must be an outcome of careful design and many ablation study, so Deepset may perform better than the baseline authors have tried in Figure 6.
> >
> > Overall, some results in appendix partly addressed my concerns on experiment but I still feel the paper can be overall improved in terms of method, problem setting, and experiment. So, I am increasing my score from 3 to 4 (although there is no score 4 in the system), in which I am still leaning toward reject.

---

### Decision · Program_Chairs · 2023-01-20

**Decision:**

Reject

**Justification For Why Not Higher Score:**

As per the above note, the main reasons include: Simplicity of task and usage of ground truth information, limiting the demonstration of the proposed method.

**Justification For Why Not Lower Score:**

N/A

**Metareview: Summary, Strengths And Weaknesses:**

This paper proposes an object manipulation task in simulation that requires learning to manipulate an object with a single distractor at training time, but then requires performing the task with a varied number of distractors at testing time. The authors propose a linearized relation network model that scales well with the number of objects in the scene. Five reviewers provided a review for this paper. One reviewer provided a positive rating but had a very short review with almost no substantial comments. As a result that review is being discounted to a certain extent. The remaining four reviewers provided a lower rating. They found the task to paper to be well written, the method to be simple and easy to reproduce, and they appreciated that the experiments demonstrate the improvements provided by the proposed model. However, they also had a few concerns. Notably, they found the task to be quite simple and insufficient to demonstrate generalization as intended. Firstly, the method uses ground truth information. Second the task does need require interacting with multiple objects or reasoning over 3 or more objects. Instead, one can solve the task by learning to ignore the distractors and only focussing on the target object. While it is good to see the value of the proposed method on this task, the reviewers felt that a demonstration on more complex suite of tasks would be more beneficial to the reader and to the community. I agree with these points made by reviewers, and hence I am recommending a reject.

**Summary Of Ac-Reviewer Meeting:**

N/A